# Brown Adipose Tissue: New Challenges for Prevention of Childhood Obesity. A Narrative Review

**DOI:** 10.3390/nu13051450

**Published:** 2021-04-24

**Authors:** Elvira Verduci, Valeria Calcaterra, Elisabetta Di Profio, Giulia Fiore, Federica Rey, Vittoria Carlotta Magenes, Carolina Federica Todisco, Stephana Carelli, Gian Vincenzo Zuccotti

**Affiliations:** 1Department of Health Sciences, University of Milan, 20146 Milan, Italy; 2Department of Pediatrics, Vittore Buzzi Children’s Hospital, University of Milan, 20154 Milan, Italy; valeria.calcaterra@unipv.it (V.C.); elisabetta.diprofio@unimi.it (E.D.P.); giulia.fiore2@studenti.unimi.it (G.F.); vittoria.magenes@unimi.it (V.C.M.); carolina.todisco@unimi.it (C.F.T.); gianvincenzo.zuccotti@unimi.it (G.V.Z.); 3Pediatric and Adolescent Unit, Department of Internal Medicine, University of Pavia, 27100 Pavia, Italy; 4Department of Animal Sciences for Health, Animal Production and Food Safety, University of Milan, 20133 Milan, Italy; 5Department of Biomedical and Clinical Sciences “L. Sacco”, University of Milan, 20157 Milan, Italy; federica.rey@unimi.it; 6Pediatric Clinical Research Center Fondazione Romeo ed Enrica Invernizzi, University of Milan, 20157 Milan, Italy

**Keywords:** brown adipose tissue, browning of white adipose tissue, childhood obesity, gut microbiota, breastfeeding, fetal programming, docosahexaenoic acid, eicosapentaenoic acid

## Abstract

Pediatric obesity remains a challenge in modern society. Recently, research has focused on the role of the brown adipose tissue (BAT) as a potential target of intervention. In this review, we revised preclinical and clinical works on factors that may promote BAT or browning of white adipose tissue (WAT) from fetal age to adolescence. Maternal lifestyle, type of breastfeeding and healthy microbiota can affect the thermogenic activity of BAT. Environmental factors such as exposure to cold or physical activity also play a role in promoting and activating BAT. Most of the evidence is preclinical, although in clinic there is some evidence on the role of omega-3 PUFAs (EPA and DHA) supplementation on BAT activation. Clinical studies are needed to dissect the early factors and their modulation to allow proper BAT development and functions and to prevent onset of childhood obesity.

## 1. Introduction

Pediatric obesity is a significant challenge for our society [1]. Prevalence between males and females, in school age and during adolescence, is better than previous years, but remains high [2]. National governments have undertaken both preventive actions (e.g., reduction of simple sugars in products for the pediatric population, traffic light labeling) [3] and therapeutic activities on lifestyle modification and dietary habits, but pediatric obesity remains a difficult disease to treat [4].

Obesity has a multifactorial pathogenesis, different factors such as genetic, psychosocial and environmental ones, and in particular chronic energy imbalance, may influence its onset [5]. One of the key roles of the adipose tissue, composed of white adipose tissue (WAT) and brown adipose tissue (BAT) in mammals, is to regulate energy metabolism through various hormonal pathways. Several systems and organs—such as the central nervous system, pancreas, liver, skeletal muscle tissue, kidneys, endothelium, immune system—are affected by the endocrine actions of the adipose tissue, thus contributing in a complex way to the homeostatic regulation of energy balance, insulin sensitivity and vascular-endothelial functions [6].

In recent years, studies are focusing on BAT as a possible therapeutic target. The purpose of this review was to examine the literature to highlight nutritional factors that play a role in the browning of WAT and possible modes of intervention acting on energy metabolism underlying the pathogenesis of childhood obesity.

## 2. Materials and Methods

### 2.1. Topic of the Review

This work is a revision of the literature published over the past 15 years and discusses the potential relationship between BAT and risk of overweight/obesity in childhood; fetal programming; BAT localization and function in adults, infants, preterms and adolescents; BAT breastfeeding and solid food introduction; BAT and gut microbiota; BAT and DHA.

### 2.2. Methodology

Publications were identified from a search in PubMed using the terms ‘brown adipose tissue’, ‘overweight’, ‘obesity’, ‘childhood’, ‘pregnancy’, ‘brown adipose tissue in infants’, ‘browning of white adipose tissue’, ‘brown adipose tissue in adolescents’, ‘brown adipose tissue in preterm’, ‘brown adipose tissue and breastfeeding’, ‘brown adipose tissue and solid food introduction’, ‘brown adipose tissue and gut microbiota’ and ‘brown adipose tissue and DHA’. Other publications came from the reference list of other papers, hand searches and from the personal reference databases of the authors.

## 3. Brown Adipose Tissue (BAT): Localization and Functions

The Brown Adipose Tissue (BAT) is a plastic tissue formed by brown adipocytes which reside in specific depots. BAT is the main site responsible for non-shivering thermogenesis (NST) in mammals whereas WAT, accumulating triglycerides, is the organ in charge of metabolic energy storage.

Brown adipocytes contain multiple small multilocular cytoplasmic lipid droplets, a central nucleus and a high number of mitochondria rich in iron that give the cell the characteristic brown color [7]. The principal characteristic of brown adipocytes is that they express high levels of uncoupling protein 1 (UCP1), specifically on the inner mitochondrial membrane. This protein serves to dissipate the energy stored in the mitochondrial electrochemical gradient as heat. This process is called non-shivering thermogenesis and it is typically used by human newborns and small mammals to maintain their core body temperature defending the body from cold [7].

In human newborns BAT depots are located mainly in the interscapular and perirenal areas [8]. At the beginning, it was widely believed that BAT rapidly declines with aging and that it does not play a physiological role in human adults. Interestingly, with some more recent studies it has been shown that BAT depots also exist in human adults and they are in the supraclavicular, paravertebral, axillar, cervical and per-aortic areas, with a high variability between individuals [8].

In these tissues, the activity of adult human BAT is responsive to temperature manipulation and it increases the expression of genes involved in lipid metabolism and their mobilization from periphery in overweight and obese men [8]. BAT also contributes to systemic clearance of glucose, stimulating its uptake by two different pathways: the sympathetic nervous system stimulation and the insulin signaling. These two pathways have a synergistic effect because both promote the translocation of glucose transporters on the plasma membrane [8]. Specifically, adrenergic stimulation by epinephrine and norepinephrine on β3-adrenoreceptors located on brown adipocytes increase the expression and translocation of glucose transporters GLUT (glucose transporter) 1 and GLUT4 to the plasma membrane. On the contrary, insulin promotes translocation of GLUT4 to the plasma membrane via the PI3K-PDK-Akt pathway [8] (Figure 1).

BAT has a distinctive secretory profile, which is quite different from that of WAT, and this is perhaps unsurprising considering that the two tissues have very different, largely opposite, physiological roles in energy metabolism. BAT depots have an important metabolic role due to the secretion of special adipokines called “BATokines”, regulatory factors (comprehensive of growth factors, cytokines, morphogenetic proteins, lipids and RNAs) with endocrine, paracrine and autocrine functions. Examples of important factors released are triiodothyronine (T3), fibroblast growth factor 21 (FGF21), interleukin-6 (IL-6), vascular-endothelial growth factor-A (VEGF-A), nitric oxide (NO) and bone morphogenetic protein (BMP) [9].

Another interesting aspect is the fact that the ability of active BAT to expend high amounts of energy has raised interest in stimulating thermogenesis therapeutically to treat metabolic diseases related to obesity and type 2 diabetes [10].

The more accepted link between BAT function in thermogenesis and protective action against obesity, hyperglycemia and hyperlipidemia should not be ascribed only to the capability of this tissue to burn glucose and lipids for thermogenesis but indeed its capacity to signal other organs and regulate systemic metabolism might also contribute to this mechanism. BAT transplantation might even be a strategy to take advantage of the beneficial effects of brown adipocytes.

Unfortunately, scientists have concluded that switching on existing BAT depots is not a viable option for treatment of obesity as the enhanced thermogenesis caused does not lead to clinically meaningful reductions in body weight [10]. An alternative approach would be to transplant new BAT derived from another individual or even better from the same donor to avoid any issue of tissue reject. BAT transplantation models have shown improvements in glucose metabolism and insulin sensitivity, as well as reductions in body mass and decreased adiposity in recipients [10].

### Brite or Beige Adipocytes

Recent data suggests the existence of another adipose cell phenotype, which shows both white and brown adipose cells features and is therefore called bright or beige adipose tissue. Beige adipocytes are usually located into WAT sites, especially in the subcutaneous WAT depots, and the phenomenon of their appearance in WAT depots is referred as “browning” [9,11,12].

Beige cells resemble brown fat cells in having multilocular lipid droplets and capacity for UCP1-mediated thermogenesis; however, at the basal, expression of UCP1 is very low [8]. In fact, the thermogenic feature of beige adipose tissue (BeAT) appears under prolonged cold exposure or as consequence of chronic β-adrenergic stimulation [9,11,12]. Under these conditions, the pre-existing beige adipocytes (which may appear unilocular in the basal state) will go through phenotypic trans-differentiation, and browning will appear both morphologically and histochemically [11]. Once stimulated beige cells activate the expression of UCP1 at very comparable levels to those of the classic brown fat cells [11]. Thus, the beige cells have the capability to switch between an energy storage and energy dissipation phenotype.

The adipogenesis of white and brown adipocytes includes the development of pre-adipocytes from mesenchymal stem cells (Myf5-negative cells or Myf5-positive cells) that further differentiate to mature adipocytes (white adipocyte or brown adipocyte respectively) [7]. Regarding the beige cells, their origins remain a matter of debate. Beige-type cells may generate from white-to-brown adipocyte trans-differentiation or transformation of white adipocytes [12]. Alternatively, it has been hypothesized that these cells origin from the de novo differentiation of a distinct sub-population of WAT progenitors (presenting CD137 and TMEM26 as surface markers) which can give rise to either white or beige adipocytes depending upon the stimuli [7,12]. Browning or “beiging” takes place in response to a variety of external stimuli such as chronic cold exposure, cancer cachexia, caloric restriction, exercise and bariatric surgery [8]. Once activated, beige adipocytes secrete multiple autocrine and paracrine factors that control the expansion and activity of BAT and the extent of browning of white adipose tissue [9]. Therefore, BeAT development and functions go along with BAT activity in the control and prevention of obesity onset.

## 4. Pregnancy and BAT in Newborns: The Role of Fetal Programming

BAT development occurs intra-utero and it is activated at the time of birth, upon cold exposure and endocrine stimulation [13,14,15]. The magnitude of activation, in terms of rise in the functioning of the mitochondrial UCP1, has been shown to be dependent on different factors, such as fetal thermal environment, type of childbirth and maternal characteristics and habits [16,17].

Given the complementary role between BAT and WAT, many studies focused on the factors involved in the development of fetal adipose tissue in general, to understand whether it is possible to modulate it. Unfortunately, the majority of the studies so far performed have been conducted in animal models that have different features compared to humans [18,19]. Here, we focus on the human studies and the consequent results, but we also cite the most relevant in vivo animal studies.

The main factors contributing to the growth of the adipose cells have been found to be strictly connected and involved in the so-called “microbiome-immune-metabolic axis”, a series of connections between the gut microbiome, the immune system and metabolic organs as adipose tissue [20].

One of the most studied contributors in fetal adiposity development is the maternal gestational weight gain. Excessive gestational weight gain, in early and mid-pregnancy, has been associated with high birth weight and lower leptin levels in offspring, factors known to increase the risk of overweight later in life [20,21,22].

Interestingly, it has also been shown that infants of mothers with an adequate gestational weight gain not only have a lower fat mass, but also a more diverse microbiome with respect to babies born from mothers with excessive gestational weight gain [22,23]. Independently from the weight gained during gestation, in humans the pre-pregnancy body mass index has been also related to an increased risk of offspring overweight [24].

The mechanisms responsible for this association have not been elucidated in humans yet. On the contrary, in mice it has been recently demonstrated that maternal obesity impairs fetal brown adipogenesis and myogenesis and moreover promotes white adiposity development through transcriptional regulation [25]. Another extremely important actor in this context is the mother’s diet during pregnancy as the nutrients are transferred to the fetus through the placenta. Maternal diet can affect the offspring in different ways [26].

First, in terms of caloric intake, a normocaloric maternal diet was found to be associated with a lower child body mass index (BMI) with respect to a high caloric dietary pattern [26]. It has also been shown that offspring of high-fed pregnant rats have significantly higher WAT and total adiposity with respect to controls [27].

Secondly, in terms of nutrient balance, a low protein and low-fat diet has been shown to enhance BAT thermogenesis and reduce birth weight in rats and modify gut microbiome increasing *Bacteroides* to *Firmicutes* ratio in toddlers [28,29].

On the contrary, in a primate model, a high-protein and high-fat diet has been found to decrease both the BAT thermogenesis and the *Bacteroides* to *Firmicutes* ratio, promoting WAT deposition in the offspring [30]. Also, in humans a maternal high-fat diet seems to influence both fetal microbiome and, later in life, the children microbiome, although the long-term metabolic effects of these findings have not been elucidated yet [31]. In rats, perinatal maternal high-fat diet has been shown to predispose to offspring obesity acting on the endocannabinoid system in both WAT and BAT. Interestingly, this system acts on both UCP1 regulating cell thermogenic activity and adrenergic signaling, two important signaling pathways in BAT (Figure 1) [32]. There are several biological mechanisms that may play a role in the influence of maternal diet on BAT development. One of the most accredited theories is the overnutrition hypothesis, which states that maternal circulating glucose can cross the placenta and lead to hyperinsulinemia in the fetus. This, in turn, promotes adipogenesis [33].

The above evidence has been confirmed by studies with metformin, one of the most used oral treatment for diabetes in pregnancy. This drug, when administered to obese pregnant mice, passes through the placenta and promotes offspring thermogenic activity, reducing WAT and enhancing UCP1 gene expression [34,35].

On the contrary, the administration of metformin to obese pregnant women seems to have no immediate effect on the adiposity of newborns, even if this does not exclude the possible protective effects later in childhood [36]. Fetal adipose tissue hypertrophy has also been linked to higher levels of maternal circulating free fatty acids and placental accumulation of triglycerides [37,38]. Another pharmacologic treatment that has been shown to modify adipose tissue growth during pregnancy is the use of antibiotics. Indeed, it has been reported that children born from mothers who take antibiotics during gestation have higher BMI, fat mass and waist circumference with respect to children whose mothers were not treated. The main culprit of this increased weight gain seems to be the microbiome [39,40,41]. As previously mentioned, the delivery method also has an impact on the BAT functioning, once more, through gut microbiome modulation [42]. Specifically, some human studies correlated the C-section with an increased risk of weight gain [43]. Interestingly, infants born through C-section have lower gut microbiome diversity, composed mainly by bacteria present on the mother’s skin. On the contrary, vaginally delivered babies host a richer microbiome, deriving from the mother’s vaginal and perianal areas [44]. Furthermore, emerging studies associated the offspring’s risk of overweight with maternal smoking, alcohol consumption and exercise, being the former two negative factors and the latter a protective one [26,45,46].

In humans, both paternal and maternal smoking during gestation, especifically in the first trimester, have been associated with an increased risk of childhood overweight. Reducing the number of cigarettes, without quitting, did not decrease this risk [47]. Mother smoking seems to affect DNA methylation; even so, the association between DNA methylation and adipocytes increase have not been elucidated in children yet, but in adults smoking-induced DNA methylation was shown to be present in adipose cells [48,49]. In mice, maternal exercise during pregnancy has been found to stimulate fetal brown and beige adipose tissue (BeAT) thermogenic activity and protect offspring from high-fat-diet-induced obesity.

Among exercise-induced hormones increased by maternal exercise, apelin was present at higher level in maternal and fetal circulation and at the level of the placenta. Moreover, upon exogenous apelin administration many BAT markers, such as UCP1, increased, suggesting that apelin itself is responsible for the exercise-related enhanced BAT activity in the offspring.

Although not confirmed in humans yet, this study is very interesting both because the high-fat-diet given to mice is extremely common in Western society and because it suggests that the apelin system could be a potential therapeutic target [50]. In mice, exercise during pregnancy has also been shown to act on the offspring hypothalamus and WAT inflammatory pathways, reducing the pro-inflammatory IL-6 and ameliorating glucose metabolism [51]. This, besides reinforcing the previously mentioned role of fetal hyperinsulinemia in adipose tissue development, underlines another factor involved in the adiposity growth: the inflammation. Taken together, all this evidence underlines the importance of maternal habits in the growth and function of the fetal adipose tissue and the possibility to modulate it, in terms of fetal programming. The idea of fetal programming is based on the so-called ‘developmental plasticity’, stating that there are specific developmental periods, such as the intra-uterine life, in which the organism adapts to environmental conditions [52,53]. As most studies concerning BAT modulation in response to specific environmental stimuli (as a high-fat diet, smoking or exercise) are animal studies [50,51,52,53], further evaluations in humans are needed, especially because BAT stimulation may be an important strategy to prevent children overweight and all the associated pathological conditions.

## 5. BAT in Term and Preterm Infants, Childhood and Adolescence

From the morphological point of view, the thermogenic fat cells that compose the adipose tissue during the pediatric period are multilocular (formed by small lipid droplets), whereas most adult adipocytes are unilocular (containing a single large lipid droplet). This structure reflects the different thermogenic activity during life [54,55].

BAT major activation occurs at birth upon cold exposure and B3 adrenergic stimulation [54,56,57]. The extra-uterine cold environment stimulates the thermogenic function of BAT, which generates heat by a process known as non-shivering thermogenesis [17,58]. This type of thermogenesis is extremely important in newborns both because their large area-to-volume ratio makes them more susceptible to heat loss and because their skeletal muscle tissue is not able to correctly maintain temperature through shivering [59]. The thermogenic activity of newborns’ BAT, especially in the interscapular region, depends on the cellular content of UCP1 and on the type II iodothyronine (DIO2) activity. Aged adipocytes instead contain less UCP1 and this is coherent with the decreased thermogenic ability of adults [13,54,60]. The main sites of BAT in infants, found first in necroscopic studies and then confirmed by non-invasive imaging tools as Positron Emission Tomography (PET) and Magnetic Resonance Imaging (MRI), are the interscapular region (the one with the highest thermogenic activity), the neck, the axillae, areas around the trachea, the esophagus and the large vessels within the mediastinum and intraabdominally in the paravertebral and perinephric spaces [54,61,62,63,64] (Figure 2). Interestingly, also the buccal fat pad is composed of BAT in the first weeks of life [65]. BAT in the buccal fat pad is necessary for newborns to develop an efficient sucking process, as it warms the masticatory muscles and produces energy. Indeed, in mice it has been demonstrated that the reduction in buccal BAT causes an impaired suckling activity [66].

At 1 month, and later in life, buccal fat pad is instead made of WAT, characterized by larger adipocytes [65]. It remains unclear whether brown cells transform into white adipocytes or the WAT develops in the empty spaces once the BAT disappeared at the level of buccal fat pad, but it has been demonstrated that in mice BAT growth at the level of the inter scapular region in the first two weeks of life is due to both proliferation and lipogenesis, while later in infancy BAT grows mainly via lipogenesis [67]. While BAT develops in embryos, BeAT is induced from WAT during postnatal life [68]. Concerning newborns, it has been hypothesized that there is a difference in terms of BAT formation between preterm or small for gestational age and appropriate for gestational age infants [69].

This was considered to be responsible for the development of cardiovascular and metabolic diseases later in childhood [52]. This has been linked to a dysregulation of the IGF-1 signaling pathway in mice [70]. In human studies performed through MRI in young adults, this correlation has not been confirmed [69,70]. Proceeding chronologically, later in infancy [71], in childhood and especially during the adolescence, BAT is involved in muscle and skeletal development [72].

It has been shown that children with active BAT have an enhanced muscle mass compared to children without identifiable BAT [73].

Moreover, brown adipocytes show many functional, molecular and histological characteristics typical of myocytes, such as the high content of mitochondria, the adrenergic response and the expression of myogenic factors [74]. Thus, these two types of cells have been suggested to share a common lineage origin [13,75]. Interestingly, in humans both muscle growth and BAT increase seem to be enhanced by physical activity [76,77].

Even if further studies are needed to confirm these observations, another common feature between brown fat cells and myocytes can be highlighted [78]. During puberty, there is both a substantial gain in muscle mass, mediated by sex hormones, and an increase in BAT that, in mice, seems to depend on growth hormone and sex steroids [79,80]. In a recent study performed in young adults, it has been demonstrated that there is a sex difference in supraclavicular BAT temperature, both in basal conditions and upon cold and meal stimulations. This was related to circulating steroid hormones. Specifically, baseline supraclavicular temperature was positively correlated with progesterone and negatively with cortisol, while cold- and meal-induced BAT temperature changes were mildly correlated with 17β-estradiol and independent from testosterone levels [81]. BAT seems to be also involved in bone maturation, as a positive correlation has been found between the number of brown adipocytes and the appendicular skeleton in children [82]. Moreover, BAT-lacking mice present a low bone mass and an increased bone resorption [83]. After seeing the physiological roles of BAT in the pediatric age, it is worth highlighting some pathological conditions that have been correlated with BAT in infants, children and adolescents. In infancy BAT has been hypothesized to be involved in the sudden infant death syndrome (SIDS) [84], as thermal stresses and dysregulated thermogenesis are involved in this condition, but this hypothesis has not been confirmed in a later research work [85].

In childhood and adolescence BAT is instead related to hibernomas, benign soft-tissue tumors mainly localized in the neck, axillae and retroperitoneum [86]. Clinically, hibernomas present as soft masses, painless and either stable or with a slow growth. These tumors mimic other lipomatous neoplasms, such as lipomas, which are more common both at the clinical level and at imaging presentation, but, deriving from brown fat, they contain more mitochondria [87,88]. The most relevant correlation between BAT and diseases remains the inverse association between brown adipocytes and cardiometabolic diseases [89,90]. There is clear evidence that in humans BAT works as a ‘glucose sink’ and it can improve glucose and insulin homeostasis through insulin-dependent and independent mechanisms. Specifically, insulin stimulated glucose uptake in BAT occurs upon membrane translocation of the glucose GLUT 4 that acts on the PI3K-Akt pathway [91]. The glucose uptake upon adrenergic stimulation is independent from GLUT4 and relies on the translocation of GLUT1 [92]. Aside for the different transporter used, the two mechanisms share similar down-stream signaling pathway [93]. BAT activation by adrenergic stimulation was demonstrated to be able to correct hypercholesterolemia and attenuate atherosclerosis in mice [94].

BAT has been also found to be involved in lipid metabolism in humans. Indeed, brown adipocyte volume was associated with enhanced lipolysis, triglyceride-free fatty acid cycling, free fatty acid oxidation and insulin sensitivity. Moreover, upon cold stimulation and increased UCP1 expression many genes involved in lipid metabolism were up-regulated in supraclavicular BAT [95].

To better understand whether and how BAT stimulation plays a role in human lipid metabolism, further human studies are needed. Interestingly, brown adipocytes surrounding the heart and vasculature may play a key role in attenuating atherosclerosis. This has been suggested by animal studies [96] and needs to be further evaluated in humans.

BAT quantity has been inversely correlated with BMI making it a possible anti-obesity target [97,98]. Recently, it has been demonstrated that higher supraclavicular BAT is correlated with a lower increase in total body fat in the first months of life [99], an indicator of increased risk of obesity later in childhood [100]. The mechanisms behind this correlation have not been clarified yet, but it has been underlined the possible role of BAT in glucose homeostasis [89]. Upon cold stimulation, glucose BAT use increases, and this improves insulin sensitivity, making BAT activation a possible therapeutic strategy against insulin resistance and diabetes [95,101]. Lastly, BAT has been correlated with autoimmune hypothyroidism in young girls, who demonstrated a reduced thermogenic response to cold stimulation in the supraclavicular region with respect to controls [102]. A positive association between autoimmunity and lower BAT activity has been observed both in animals and in humans [103]. This raises the possibility that a decreased thermogenic ability of brown fat cells may predispose to the onset of an immune dysregulation. To conclude, BAT is a very active and dynamic entity during the pediatric life period, and it changes as children grow. During the neonatal period BAT is fundamental for NST [59], in childhood and adolescence BAT has a key role in skeletal and muscular development [73], later in life and during adulthood it seems involved in important metabolic processes such as glucose and lipid metabolism [95,103].

As it is part of all these biological processes, BAT affects the individual’s health at different levels (metabolic, structural, functional), thus, it is fundamental to understand whether and how we can modulate it during childhood to obtain health benefits in future. Further studies, especially in the pediatric population, are needed to draw conclusions concerning the actions of brown fat cells on lipid metabolism and its possible role as protective factor against the development of metabolic and cardiovascular disorders. (see Figure 2).

## 6. BAT Modulation

### 6.1. Breastfeeding and Solid Foods Introduction

Breastfeeding has been shown to have a role in adipose tissue modulation and, consequently, on development of overweight or obesity during childhood [104]. Indeed, breastfeeding was shown to reduce the incidence of overweight and obesity in children who were breastfed for six months. The latest COSI report reported higher prevalence of obesity among children who had never been breastfed and/or had been breastfed less than six months compared to those who were breastfed for more than six months [105]. Exclusive breastfeeding for more than 6 months was shown to lower the risk of obesity at 6 years, particularly in children with high born weight [106].

During breastfeeding, infants control the amount of milk ingested and feed according to their hunger, to develop healthy eating habits [107]. Breastfeeding also shows an important influence in the development of a healthy gut microbiome, which plays an important role in the prevention of various kinds of illnesses, including obesity [42].

The gut microbiome communities significantly vary between infants who were exclusively breastfed, and the ones fed with milk formula, or a mix of breastmilk and formula. Specifically, an abundance of genus *Bifidobacteria* was found in the microbiome of breastfed infants, compared with formula-fed infant and combination feeding infant at 6-week-old [106]. A reduced count of *Bifidobacteria* in infancy is related with a greater risk of being overweight at 10 years [108]. Breastfed infants showed increased levels of UCP1 and adipocytes characterized by abundant multilocular lipid droplets and UCP1-enriched mitochondria. This is consistent with the widespread presence of BeAT in inguinal adipose tissue (iAT) [109]. Conversely, infant who were not completely or rarely breastfed do not present these characteristics. Breast milk is highly enriched in alkyglicerols (AKGs), secreted from mammary gland cells. AKGs promotes BeAT activation, increasing the expression of BeAT-related genes and the number of mitochondria in iAT. In response to AKGs, adipose tissue macrophages of primary humans and mice produce platelet-activating factor (PAF). PAF is a powerful mediator of the immune inflammatory response and a phospholipid activator [110]. The activation of PAF receptor stimulates IL-6 production, which acts on the pro-adipogenic JAK/STAT/TGF-BETA/SMAD3 signaling pathway in AT progenitor cells and promotes beige adipocyte differentiation in iAT [111]. As previously mentioned, also in this context there is a strong correlation between BAT and the immune system.

AGKs therefore have a role in the maintenance and differentiation of BeAT during infancy. Thus, it has been suggested to supplement formula milk with AKGS for infants who are not breastfed (and therefore lack AGKs), to prevent childhood obesity [110]. This is supported by experimental evidence on a mouse model, where AGKs supplementation in adult obese mice stimulates the development of BeAT. On the other hand, no similar effect has been observed in lean mice [109]. It is interesting to notice that maternal high-fat diet (HFD) increases the risk of many diseases, such as cardiovascular disorders, hypertension, insulin resistance and type 2 diabetes mellitus in the offspring adulthood [111,112,113,114,115]. Moreover, in mice, maternal HFD during lactation was correlated with harmful metabolic responses to HFD consumption in offspring later in life, such as adipocyte dysfunction. At weaning, BAT thermogenesis is compromised and later in life BAT thermogenic activity and the UCP1 expression is lowered by maternal HFD during lactation [116].

Lastly, maternal polar lipids—enriched milk fat globule membrane (MFGM-PL) has been shown to have a promising effect in promoting thermogenesis in male offspring of rat dams during pregnancy and lactation [117]. This effect is mediated by the promotion of BAT function and browning of inguinal WAT. In turn, this increased energy expenditure alleviates obesity thus protecting against maternal HDF-induced obesity in offspring [118].

Interestingly, also solid food introduction before 4-months of age has been correlated with a greater risk of obesity at 12 months [119,120].

Moreover, early complementary feeding is related to an increased risk of overweight and obesity during adulthood in rats, through the reduction of the in vivo BAT sympathetic nervous system activity [121]. Importantly, BAT modulation shows some significant differences in males and females, likely because estrogens, as seen above, can reduce BAT thermogenesis [79,122]. In turn, BAT thermogenesis has a well understood protective effect against obesity [123,124]. Although early introduction of food is associated with obesity risk in the mice model, there are currently no studies linking early introduction of solid food and BAT activity in animals or humans.

### 6.2. Gut Microbiota and BAT

Gut microbiota is considered an organ that lives in symbiotic relationship with its host. Its changes occur since early infant life and are determinant for the maturation of immune and metabolic functions [28,31,125]. Many studies have pointed out the role of gut microbiota as an important modulator of host homeostasis and energy balance, in fact gut microbes affect energy harvest from the diet and consequently energy storage, including body fat content [117,126]. Therefore, the identification of a possible manner of enhancing browning of WAT and thermogenic pathways passes potentially through gut microbiota activity [117].

### 6.3. Environmental Factors Affecting Gut Microbiota and BAT

Originally, animal studies were carried out to better understand the relationship between BAT and gut microbiota. Mestdagh et al. studied the modulation of BAT lipid metabolism in the absence of gut microbiota, i.e., in germ-free (GF) mice, in which the missing activity of microbes stimulates both hepatic and BAT lipolysis (through β-oxidation) while inhibiting hepatic lipogenesis [127]. In line with this study, the depletion of gut microbiota, either by means of antibiotic treatment or in GF mice, promotes the rising of brown fat cells in the inguinal subcutaneous adipose tissue and perigonadal visceral adipose tissue, together with increased expression of brown fat markers. These metabolic improvements are mediated by enhanced type 2 cytokine signaling in WAT depots and are reversed under recolonization of gut with microbes [125]. Conversely, a recent study considered the role of gut microbiota depletion on browning of WAT, referring to this condition as a trigger that negatively affects the adaptative thermogenic capacity of BAT and WAT in mice. In fact, in this study gut depletion impairs UCP1-dependent thermogenesis capacity compared to healthy gut microbiota controls [128]. The microbiota-fat signaling axis might be the reason of conflicting responses to microbiota depletion and BAT activation, the same as gut metabolites, whose effects are later examined in this paper. BAT thermogenesis is responsive to microbiota composition; therefore, an intact and healthy microbiota may be an essential component of thermoregulatory response and with this in mind, more studies still must be performed [128].

As well as antibiotic treatment, cold exposure and fasting or caloric restriction are also situations that modulate the relationship between gut microbiota and BAT [117]. The former is responsible for a dramatic change of microbiota composition. Cold exposure of mice promotes a shift in phyla proportions increases *Firmicutes* abundance over *Bacteroidetes*, enhances *Firmicutes* richness, decreases *Verrucomicrobia* phyla, particularly *Akkermansia muciniphilia* species. The expansion of this “cold” microbiota favors mice tolerance to cold, increases energy expenditure, as well as lowering fat content in part mediated by browning of white fat depots [129]. Supporting the role of cold exposure, cold transplanted mice had reduced fat mass and adiposity, as well as increased expression of UCP1 mRNA and BAT markers compared to controls. Moreover, in mice, the ambient temperature reduction protects the host from diet-induced obesity, due to the strong and rapid induction of the brown adipose phenotype under cold exposure [130]. Therefore cold-associated gut microbiota could induce BAT activity or enhance browning of white fat depots [129,130].

Consistently, also intermittent fasting or caloric restriction promotes a microbiota shift in mice boosting *Firmicutes* while decreasing most other phyla, which results in an increased *Firmicutes Bacteroidetes* ratio [117,131].

The relationship between fasting and gut microbiota was found transplanting microbiota from intermittent fasting mice to microbiota depleted mice. This intervention enhanced browning of WAT in the latter. These findings suggest that fasting condition can induce beige condition in subcutaneous inguinal WAT by enhancing UCP1 expression, while BAT seems not to be activated when exposed to same conditions. Moreover, data revealed that fasting primarily alters gut microbiota composition to promote particularly the generation of acetate and lactate and, subsequently, induces selective activation of beige WAT in mice. Therefore, an intermittent fasting treatment promotes adaptative NST and browning of subcutaneous WAT by shaping gut microbiota [131]. In addition, a more recent study investigated how caloric restriction can impact gut microbiota composition to promote browning of WAT in mice. In line with previous studies mice transplanted with microbiota from calories-restricted donors showed a decreased weight gain and an enhanced expression of brown fat markers in subcutaneous and visceral WAT [132].

Together these findings underline the importance of microbiota-immune system-fat signaling axis and strongly suggest that alterations in gut microbiome are a key factor that modulate energy uptake and whole-body energy demand with large therapeutic potential [117,132,133].

### 6.4. Modulating Factors of Gut Microbiota

Given that diets have an impact on composition of gut microbiota, it is logical to assume that manipulation of gut microbiota (e.g., through the administration of prebiotics, probiotics and postbiotics) could potentially be a browning agent and BAT activator, fundamental for the treatment or prevention of diet-induced obesity [126,134,135]. The complicated interactions among gut microbiota and BAT activation remain partially unresolved, necessitating more studies, especially on human.

#### 6.4.1. Prebiotics

Prebiotics and probiotics must be taken into account as microbiota modulators and preventive agents against gut dysbiosis and obesity-related metabolic disorders. Prebiotics consist of selectively fermented dietary ingredients and non-viable food components, which result in alterations in the composition or activity of the gut microbiota, with beneficial effects on host health [136,137].

Resveratrol is a natural polyphenol and growing evidence has indicated that it likely attenuates obesity by primarily reshaping the gut microbiota [138,139]. Serrano et al. demonstrated how resveratrol supplementation can limit weight gain toward browning of subcutaneous WAT in newborn mice through lactation, which is a critical period for metabolic programming and shaping of microbiota [140]. Resveratrol prevents the increase in *Firmicutes*/*Bacteroidetes* ratio, which is the hallmark of obesity-driven dysbiosis by decreasing relative abundance of genus *Lactobacillus* in mice [141]. Therefore, the direct action of resveratrol as prebiotic enhancer of BAT activity and WAT browning is proven to be gut-mediated and transferable with transplantation in mice, underling that the beneficial effects are transferable and gut-mediated [141].

Moreover, the class of anthocyanins, specifically the supplementation with Cyanidin-3-glucoside (C3G), have proven to ameliorate obesity profile by up-regulating BAT mitochondrial function and beige formation in subcutaneous WAT of mice [142]. A recent study demonstrated that C3G is a potential prebiotic which mitigates high-fat–high-sucrose-diet-induced disorders and gut dysbiosis in mice, by increasing relative abundance of *Bacteroidetes* phylum and expanding *Muribaculaceae* family bacteria [143].

In addition, prebiotic activity of camu camu (Myrciaria dubia), a mixture of phenolic compounds, increases energy expenditure and up-regulates UCP1 in BAT and WAT, preventing against fat accumulation. These effects are proven to be totally dependent on gut microbiota activity after transplantation of microbiota from treated mice to germ-free mice and the major contributor of thermogenesis boosting was *Akkermansia muciniphila* [144].

Another extract rich in polyphenols is green tea leaves extract (GTE), which has proven successful in reducing high-fat-induced adiposity in mice in both WAT and BAT. In fact, GTE induces browning process in WAT while counteracts whitening process in BAT and those effects suggest a thermogenesis induction together with an anti-obesity effect of green tea extract [145]. In particular, several recent animal studies have demonstrated that GTE act as a healthy prebiotic modulating gut microbiota and encouraging many species including *Akkermansia muciniphila* [146,147,148].

Furthermore, high esterified pectins (HEP) are an interesting example of prebiotic that can protect from fat accumulation by increasing energy expenditure by means of a greater thermogenic capacity and browning in BAT and WAT, respectively. Thus, HEP promote an increase in selected beneficial bacteria in the gut, in particular of acetate producers such as *Bifidobacterium*, which in turn affect energy metabolism and thermogenic capacity of rats [149]. Moreover, Dewful et al. focus on the role of inulin impact on gut microbiota composition which correlated with a slight decrease in fat mass and with metabolic changes in obese women. The selective modulation drive by inulin promoted the rise of *Firmicutes* and *Actinobacteria* over *Bacteroidetes* at the phylum level, while interestingly specific changes at genus level show an increase of *Bifidobacterium* [150]. Another trial on obese children under oligofructose-enriched inulin supplementation confirmed that prebiotic consumption normalizes childhood weight gain and reduces whole-body and trunk fat while affecting microbiota composition [151]. Oligofructose consumption thus resulted in a significant bifidogenic response by increasing *Bifidobacterium* spp. abundance, particularly enhancing *Bifidobacterium longum* and *Bifidobacterium adolescentis* while inhibiting *Bacteroides vulgatus* and microbial metabolites, such as fecal bile acids, are one of the many potential mechanism through which changes in gut microbiota impact host physiology [151]. By the way, the role of prebiotic inulin-derived fiber on body weight and metabolism has still to be elucidated, since data on humans are still controversial [134,150,151].

#### 6.4.2. Probiotics

Bearing in mind the relationship between gut microbiota and host health metabolism, the activity of probiotics on BAT has recently been investigated in animal models. Probiotics are “live microorganisms that confer a health benefit when consumed in adequate amounts” [137]. Mainly *Lactobacillus* plantarum has proven successful in potentiating process of BAT thermogenesis via activating UCP1-dependent mechanisms when administrated in rats, and these results suggested that *L.plantarum* may thus inhibit the progression of obesity in animal models [152,153]. In line with this study also *Lactobacillus amylovorus* administration exerts browning activity on mice WAT, and this effect can be attributed to both increasing PGC-1α expression and to remodeling of the PPAR-γ transcriptional complex, which are favorable for the expression of BAT-specific genes, such as UCP1 [154]. Considering these findings, probiotics might be an efficient therapy in modulating adaptative thermogenesis, and consequently obesity-induced phenotype in mice, but more studies still must be done.

Probiotics are a useful strategy in the control of human obesity phenotype [136,155,156], for example maternal supplementation of *Lactobacillus rhamnosus* determines a lower weight gain in the first year of life acting positively on children gut microbiota shaping [157]. Consistent with these findings, children are not the only ones benefiting from a reduction in BMI after probiotic administration [158,159], but also adults, in which for example *Lactobacillus gasseri* administration reduces body weight and abdominal adiposity [160]. Similar results were found after treatment with a combination of different strains of *Lactobacillus* (mainly *L.acidophilus*, *L.brevis*, *L.casei* and *L.salivarius*) and *Bifidobacterium* (particularly *B.bifidum* and *B.lactis*) in obese women, leading to a reduction in visceral fat and waist circumference [161]. Even if several human clinical studies outline the role of probiotics as obesity-therapy, none of them have considered BAT modulation as a possible hidden mechanism [136,155,156].

#### 6.4.3. Postbiotics

Considering postbiotics as “soluble factors secreted by live bacteria or released after bacterial lysis, which confer physiological benefits to the host” [136], many studies have investigated microbiota metabolites, produced from undigested material, known as short-chain fatty acids (SCFAs). SCFAs, namely butyrate, acetate and propionate, are gut metabolic end-products that directly modulate host health and affect differently the metabolism, especially that of adipose tissue. In fact, it has been consistently reported that the increase in energy expenditure stimulated by SCFA is associated with a whole-body lipid oxidation, passing also through an increase in BAT activity [162,163]. Gao et al. reported the role of butyrate as preventive against diet-induced obesity in mice thanks to an increase of both energy expenditure and lipid oxidation with an enhancement of thermogenesis and UCP1 genes expression [164]. In addition, acetate administration also induces browning of subcutaneous adipose tissue and leads to a contraction of body adiposity under high-fat diet in mice [165].

There is thus a potential relationship among gene expression markers of BAT and gut microbiota that was assessed in morbidly adult obese subjects, where the role of *Firmicutes* relative abundance (RA) are positively associates with the increase of browning markers (SAT DIO2, SAT PRDM16 and UCP1 mRNAs) SAT, possibly through circulating acetate [166]. They found that within the family from *Firmicutes* phylum, the bacteria from *Ruminococcaceae* family were positively correlated with insulin sensitivity, plasma acetate and PRDM16 mRNA levels. Previously, it has been demonstrated in mice that an increase of energy harvest between cold exposure led to a radical shift, increasing *Firmicutes* versus *Bacteroidetes* RA together with browning of WAT. This change in the gut microbiota is driven by increased intestinal absorptive area and an improved ability to extract energy [129]. Canfora et al. have demonstrated that acetate, propionate and butyrate colonic infusions increased fasting fat oxidation and resting energy expenditure in overweight and obese men [162]. Interestingly, some studies reported that resting energy expenditure (REE) is higher in obese children and adolescents with respect to lean subjects when expressed in absolute terms (kcal/day). Indeed, free fat mass (FFM) expressed as kg/total weight and energy intake were higher in those individuals [167,168]. Until now, no metabolism alterations have been demonstrated in obese children, so the missing link could be the *Firmicutes* and *Bacteroidetes* ratio that when compromised could cause the aforementioned events with a double impact: the positive effect from gut microbiota on BAT but at the same time, a greater intestinal absorptive area and ability to extract energy.

Recently, some studies have pointed out the possible role of bile acids as postbiotics in the modulation of BAT, and for example oral consumption of chenodeoxycholic acid (CDCA) increases BAT activity and UCP1 expression in healthy women [169]. Indeed, bile acids metabolism depends on gut microbiota, which deconjugates them, therefore when gut microbiota alteration occurs bile acid profile is also influenced. Changes in mice gut microbiota in response to cold exposure increases levels of conjugated bile acids, positively affecting signaling pathways connected to thermogenesis such as the inhibition of nuclear receptor farnesoid X receptor (FXR) and the activation of G-coupled receptor TGR5 [130,135,170]. Thus, the mechanism by which bile acids influence browning of adipose tissue represents a postbiotic challenge that still needs to be better elucidated [135].

### 6.5. Limitations of BAT Activation

The limits of BAT activation, in conclusion, are represented by environmental factors but also by physiological factors. It has been reported how some characteristics of the mother during breastfeeding may limit the activation of BAT. These are represented by the nutritional status of the mother [25], her diet excessively rich in fat during breastfeeding [28,29], and the type of breastfeeding. Formula milk does not contain AGKs that promote the activation of BAT [110,111] and, moreover, a too early complementary feeding shows a reduction of the vivo BAT sympathetic nervous system activity, and may represent factors that limit the activation of BAT [121]. In addition, a central role is given to the gut microbiota since the use of antibiotics generating a microbial depletion limits BAT activation [130]. Environmental factors such as the continuous exposure to mild temperatures determines a lower activation of BAT with consequences on glucose homeostasis [91]. Moreover, specific physiological factors reduce the activation of BAT and these are essentially represented by the state of maturation of the organism. During the growth of the child the amount of BAT present in the organism decreases. In childhood BAT is present in greater quantities in the early stages of life, promoting NST, then it begins to decrease, but it plays an important role in the development of the skeletal muscle system and later in adulthood it is involved in lipid and glucose metabolic processes [72]. Therefore, the “maturation” of the body, physiologically determines the decrease of BAT amount limiting its activation. Consequently, if BAT activation is properly promoted in childhood, it would allow the body to face adulthood with structural and metabolic functions at their best.

## 7. Studies on Functional Nutrients on BAT: The Role of Omega-3 Fatty Acids

Docosahexaenoic acid (DHA, C22:6n-3) and eicosapentaenoic acid (EPA C20:5 *n*-3) are two omega-3 polyunsaturated fatty acids. Along with α-linolenic acid (ALA, C18:3 *n*-3) and acid linoleic (LA 18:2), it is necessary to introduce them with the diet since in mammals they cannot be synthesized de novo. Although ALA and LA are precursors of EPA and DHA, their bioconversion is low (8–12% EPA, <1% DHA) [171]. The sources of this nutrient are represented only by fatty fish (salmon, trout, tuna, anchovies, mackerel), while in the marine world it is mainly present in phytoplankton [172]. The Western diet, richer in omega-6 fatty acids, determines an unbalanced omega-6: omega-3 ratio towards the omega 6 series [173]. The beneficial effects of omega-3 PUFA have been widely demonstrated for both cardiovascular diseases, weight loss and metabolic disorders [173]. Omega-3 fatty acids also have effects on adipose tissue (both WAT and BAT) and on the regulation of its metabolism, specifically on adipogenesis [173].

### 7.1. Effect of Omega-3 on WAT Function and Browning 

Omega-3 (EPA and DHA) can potentially exert an effect on the well-known obesity phenotype, characterized by a chronic low-grade inflammation, in which PUFAs might mediate some metabolic and pro-inflammatory mechanisms [174,175]. Omega-3 are key components for the regulation of WAT, whose inflammation, lipid metabolism and adipogenesis are linked directly to PUFAs activities.

Increasing evidence underlines that omega-3 PUFA might have a role on adipogenesis and lipid metabolism of WAT in animal models. Specifically, daily administration of omega-3 (460 mg DHA + 130 mg EPA) equal to 15% of total dietary lipids, together with a mild caloric restriction, synergistically induce mitochondrial fatty acid oxidation in WAT, resulting in increased oxidation of metabolic fuels and lipid catabolism in male mice [176]. Similarly, Le Mieux et al. showed that supplementation with EPA in a high-fat diet in mice resulted not only in increased mitochondrial oxidation, but also influenced adipose cellularity, which means decreased adipocyte size via reduction of lipogenesis and hypertrophy of WAT [177]. Consistent with these results, recent studies demonstrated that fish oil intake delayed weight gain and lowered lipid accumulation in WAT of mice fed with a high-fat diet [178,179].

Studying the beneficial effects of omega-3, different works pointed out that the greater is the degree of individual’s adipose inflammation the higher the omega-3 effects are [180], and indeed subjects with the highest number of WAT macrophages exhibited the highest reduction of those under 12-week fish oil capsule administration (4 g/day of EPA + DHA) [181]. Consistent with these observations, 6 weeks supplementation with omega-3 (3600 mg/day of EPA + DHA), under evoked endotoxemia condition in healthy individuals, resulted in the down-regulation of immune-related genes in adipose tissue. Thus, omega-3 administration seems to potentially alter the systemic inflammatory response influencing the secretion of immune-related signaling factors in human’s adipose tissue [182]. Furthermore, omega-3 down-regulate WAT inflammatory gene expression, such as TNF-alfa and MCP-1, under 4 g a day administration of EPA + DHA or of exclusively DHA [180,181,182,183]. Moreover, it was observed that a dietary regimen based on the consumption of omega-3 rich foods (mainly fish products) affects adipokines profile increasing adiponectin and suppressing resistin [184,185]. Conversely, Hames et al. attempted to demonstrate beneficial effects of omega-3 administration (3.9 g/day EPA + DHA) without succeeding, as insulin-resistant obese adults had no beneficial effect on lipolysis or adipose tissue inflammation after 6 months of supplementation [186]. Future studies correlating insulin resistance, WAT disfunction and omega-3 intake are necessary to better understand hidden modulating mechanisms.

Affecting adipogenesis also means impacting on WAT signaling pathways, therefore many studies pointed out the effect of omega-3 PUFAs on leptin production [187,188]. For example, 1.3 g/die of EPA supplementation in obese women has proven successful in preventing the fall of leptin levels during weight loss programs [187]. Conversely, omega-3 on lean individuals exert the opposite effect, i.e., PUFA administration of marine origin moderately decreased serum leptin [188]. Given these findings, additional studies are necessary to better understand omega-3 effects on adipokines and adipose pathway in WAT of obese and non-obese individuals.

Not only omega-3 PUFAs per se, but also the ratio between omega-6: omega-3 found in human diet might impact lipogenesis and adipogenesis of WAT. This assumption is supported by studies on breastfed infants, whom growth and development are directly linked to human milk fatty acid composition, which is primarily reflective of maternal diet and BMI [189]. Rudolph et al. demonstrated that infant breastfed with higher ratio *n*-6: *n*-3 have greater fat mass compared with those fed with lower ratio values [190]. Thus, a high human milk *n*-6: *n*-3 fatty acid ratio may significantly contribute to infant adipose deposition during early life nutrition [190].

As well as affecting functionality and regulation of WAT, omega-3 PUFAs might influence the browning of this tissue under specific condition. In vitro studies suggest that EPA (100–200 μM for 24 h) could trigger the interconversion of subcutaneous white adipocyte into a beige-like phenotype, stimulating those routes involved in thermogenic capacity, mitochondrial biogenesis, fatty acid oxidation and UCP1 gene expression [191,192]. Consistent with these studies, fish oil intake increases beige adipocyte markers in inguinal WAT by up-regulating both UCP1 and β-3 adrenergic receptor [193]. The β-3 adrenergic receptor (β-3AR) activation contributes to thermoregulation and energy homeostasis via sympathetic stimulation of BAT adaptive thermogenesis. Moreover, other data indicate that the dietary omega-6: omega-3 ratio is essential for beige adipocyte recruitment via β-3AR stimulation [194]. Omega-3 PUFAs increase browning in the context of obesity by several possible ways, for example through oxilipins, which are PUFA derivatives that might affect positively or negatively browning of adipocyte whereas they are of omega-3 or omega-6 origin, respectively [174,195]. In this prospective adjusting the omega-6 and omega-3 dietary ratio could work as major regulator of adaptative thermogenesis involved into brown and beige adipocyte recruitment [194].

As reported previously a possible mechanism of WAT browning lies into the gut microbiota activity and this effect can be mediated thanks to the type of dietary lipids. In fact, gut microbiota exacerbates metabolic inflammation through increased Toll-like receptors in the systemic circulation and WAT inflammation in mice fed with lard diet compared to mice fed with fish oil [196]. Thus, microbial components are involved in the mediation of the inflammatory and metabolic phenotype depending on dietary lipids, but further studies are necessary.

The complexity of the hidden mechanisms by which omega-3 affect WAT function and browning emphasizes the need to increase our current understandings of their adipogenic properties to modulate energy balance.

### 7.2. Effect of Omega-3 on BAT Function

Various physiological conditions may influence BAT activity in humans: cold, circadian rhythms, and exercise [197]. In mice, it has been demonstrated that a dietary supplementation for 8 weeks of 3 g/body weight kg of CLA and/or Fish Oil (FO), composed by 64% EPA and 28% DHA, can change body metabolism (higher body energy expenditure) together with mitochondrial functions in several organs including BAT [198,199]. Even so, PUFAs are a very varied class of fatty acids, including both omega-3 and omega-6, with contrasting effects on BAT. In fact, what seems to be crucial in promoting thermogenesis in BAT is the omega 3 to omega 6 ratio [194]. EPA and DHA can decrease fat accumulation and promote thermogenesis in both brown and beige adipocytes. The mechanism by which EPA and DHA promote the activation of BAT and induce the browning of WAT has been demonstrated in fish oil fed mice finding an increase in basal temperature and oxygen consumption [175]. Also in vitro it has been found that EPA improves the respiratory capacity of brown adipocytes, content of mitochondria, glycolytic capacity and increased maximal, basal and uncoupled respiration [175]. This mechanism is summarized in Figure 3.

In humans, several studies have shown that fish oil supplementation does not reduce body weight, but some studies indicate that it may have effects in reducing fat mass. In children, however, it seems to attenuate the accumulation of adipose tissue during growth [192], while in young normal-weight adults a supplementation of 6 g/day of fish oil for three weeks increased fat oxidation and decreased fat mass [200,201,202]. Finally, although several meta-analyses [202,203] also show a reduction in waist circumference following fish oil supplementation, the effect by which fish oil promotes fat mass loss, primarily from visceral deposits and activation of human BAT, remains not completely dissected [204].

Recently, associations between BAT and circulating levels of certain lipid species (classes of non-esterified fatty acids, NEFAs, and oxylipin lipids) have been identified in mice [205]. In humans, a decrease in weight gain and an increase in BAT activity following high dietary omega-3 intake (from fish or fish oil supplementation recorded by validated semiquantitative food frequency questionnaire) has been observed only in genetically driven, high-number cohort studies and not in clinical trials [206]. A direct mechanistic role of EPA and DHA in the up-regulation of BAT activity in humans to date is supported by preclinical studies [175,193,194,207,208,209]; however, possible biomarkers of BAT activity could be the indirect relationships as a result of NEFA oxidation [205]. To fully elucidate the relationship between omega-3 NEFAs and BAT activity, further human studies are needed, including NEFA interventions and tracers.

In conclusion, EPA and DHA have been shown to increase thermogenic activity in mice by activating the expression of UCP1 and miRNAs. Some epigenetic factors could also underlie the activation of BAT by omega 3. In humans there are conflicting results but, mostly, it is not yet clear how EPA and DHA activate BAT. Finally, NEFA oxidation need to be investigated in humans to clarify if they can be evaluated as potential biomarkers of BAT activity.

## 8. Conclusions

It is well known that obesity has a multifactorial pathogenesis [5]. In recent years, research has been focusing on BAT as a possible therapeutic target to influence energy metabolism. BAT is a plastic tissue formed by brown adipocytes which reside in specific depots, it contributes to both glucose and lipid metabolism [90,91,92], but also has endocrine, paracrine and autocrine functions (see Figure 2). BAT develops in the early stages of life and then declines during growth [8]. In newborns, it is found mainly in the interscapular and perirenal areas as well as in the buccal pad during the first two weeks of life [54]. In preterms or small for gestational age there is a difference in BAT formation compared to appropriate for gestational age infants, probably due to an IGF-1 signaling pathway dysregulation [69]. The other main sites are supraclavicular, paravertebral, axillar, cervical and per-aortic areas [8]. Some factors may influence the proper development of BAT. In the early stages of life exposure to cold is crucial as well as physical activity and sex hormones during adolescence [82] (see Figure 2). Recently, BAT has also been found in adults [10].

The habits and health of the mother can play a part in the subsequent development of BAT in the newborn. In fact, mainly from animal models studies (as summarized in Figure 4), it is clear that an excessive pre-pregnancy and during pregnancy BMI, a maternal diet rich in fat and calories, together with high levels of circulating glucose and free fatty acids can promote adipogenesis in the fetus [26,27,28,29,30]. The maternal diet has an important role also during breastfeeding as if it is rich in fat it seems to reduce the thermogenic activity of BAT and the expression of UCP1 in newborn mice [50]. Instead, breastfeeding mice showed a better expression of UCP1 and may benefit from both AKGs and MFGM-PL promoting BeAT activation in iAT [110]. Complementary feeding is another important step because it appears that early introduction of solid food before 4 months of age not only increases the risk of overweight and obesity but also reduces the activity of BAT by the sympathetic nervous system [119].

Furthermore, an aspect that covers all pediatric stages is the health of the microbiota. Although evidence is conflicting, the one piece that seems to play a central role is the Firmicutes Bacteroidetes ratio [127,133,136]. Indeed, certain factors (such as exposure to cold, fasting or caloric restriction, antibiotic use) affect the activity and proper development of BAT via the gut microbiota [127,130,132]. A potential therapeutic intervention in obese children could be the modulation of the microbiota through the administration of prebiotics, probiotics, and postbiotics, although most of the studies conducted so far have been performed in mice.

Omega 3 fatty acids also promote both white and BAT and have a role in promoting adipogenesis [174]. EPA and DHA have been shown to increase thermogenic activity in mice by activating the expression of UCP1 and several miRNAs [195]. Some epigenetic factors could also underlie the activation of BAT by omega-3 [190]. In humans there are conflicting results that showed a reduction in waist circumference and accumulation of adipose tissue during growth, but the mechanism remains unclear [205]. NEFAs could be possible biomarkers of BAT activity [206], but several humans studies are needed to fully elucidate associations between BAT and NEFAs circulating levels.

In conclusion, more human studies are needed to assess early factors (from pregnancy to early life) that may modulate BAT activity, to possibly justify clinical studies on this topic.

## Figures and Tables

**Figure 1 nutrients-13-01450-f001:**
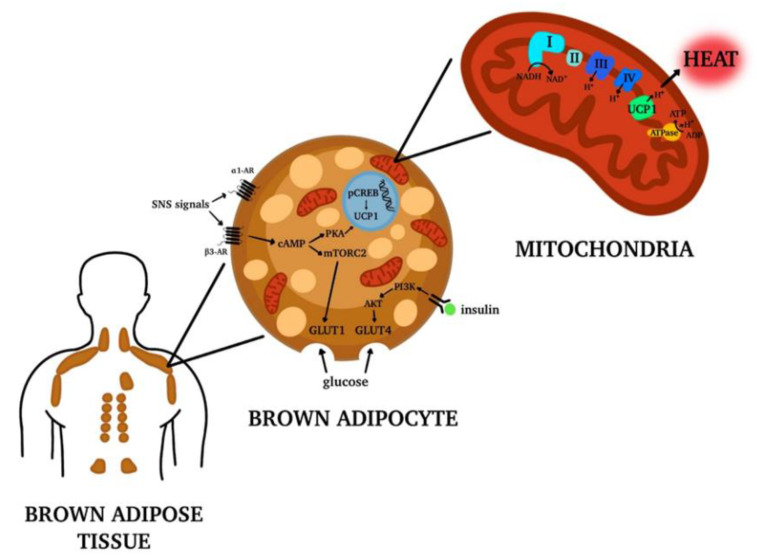
Brown Adipose Tissue (BAT) distribution in adult human body. In adult humans of BAT depots are located mainly in the supraclavicular, paravertebral, axillar, cervical and per-aortic areas. The tissue is formed by brown adipocytes, characterized by multiple lipid droplets and the expression of high levels of uncoupling protein 1 (UCP1) on the inner mitochondrial membrane. UCP1 is responsible for the release of energy in the form of heat, generating the process called non-shivering thermogenesis. Moreover, the brown adipocytes activation contributes to systemic clearance of glucose and lipids. Made by © BioRender 2021.

**Figure 2 nutrients-13-01450-f002:**
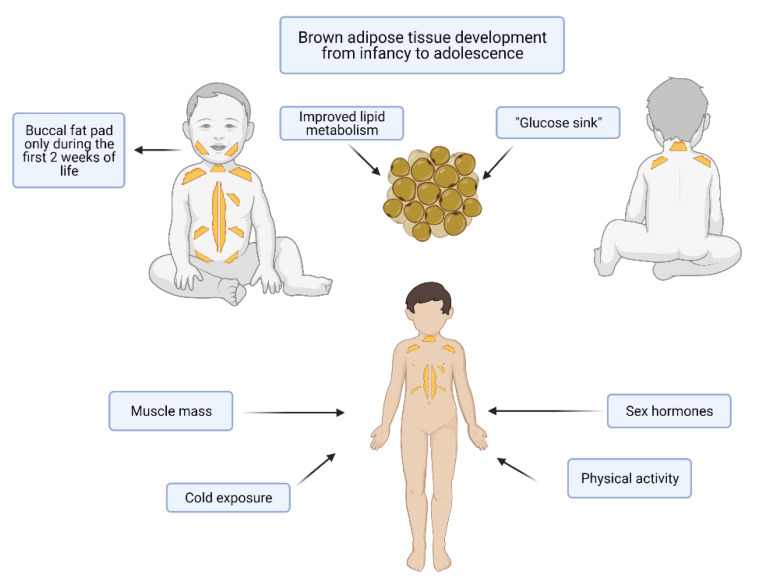
Brown adipose tissue (BAT) in infants and adolescents. The main sites of BAT accumulation in infants are the interscapular region (the one with the highest thermogenic activity), the neck, the axillae, areas around the trachea, the esophagus and the large vessels within the mediastinum and intraabdominally in the paravertebral and perinephric spaces. The buccal fat pad is composed of BAT in the first weeks of life. During puberty, there is both a substantial gain in muscle mass, mediated by sex hormones, cold exposure and physical activity. Made by © BioRender 2021.

**Figure 3 nutrients-13-01450-f003:**
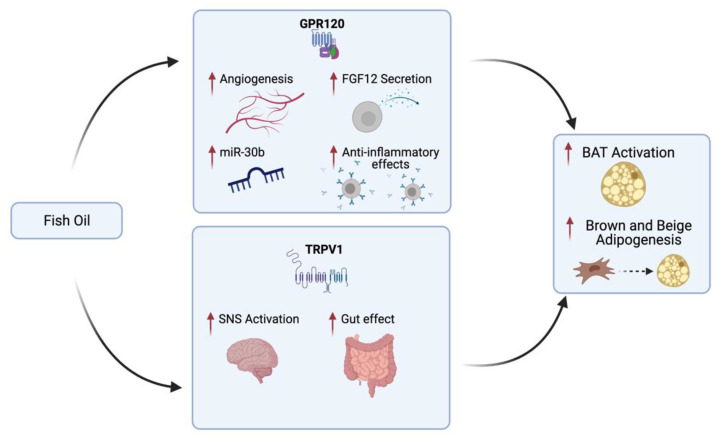
Fish oil mechanism promoting brown adipose tissue (BAT) activation and brown and beige adipogenesis in vivo. Fish oil activates TRPV1 in the gastrointestinal tract and consequently stimulates the sympathetic nerves that innervate the fat cells, causing adipose thermogenesis through β2-adrenoreceptors. Furthermore, fish oil acting as a ligand of GPR120 induces several effects on adipocytes such as secretion of FGF21 and VEGF-A (which promotes angiogenesis and anti-inflammatory effects through activation of immune cells) and expression of miR-30b [200]. Made by © BioRender 2021.

**Figure 4 nutrients-13-01450-f004:**
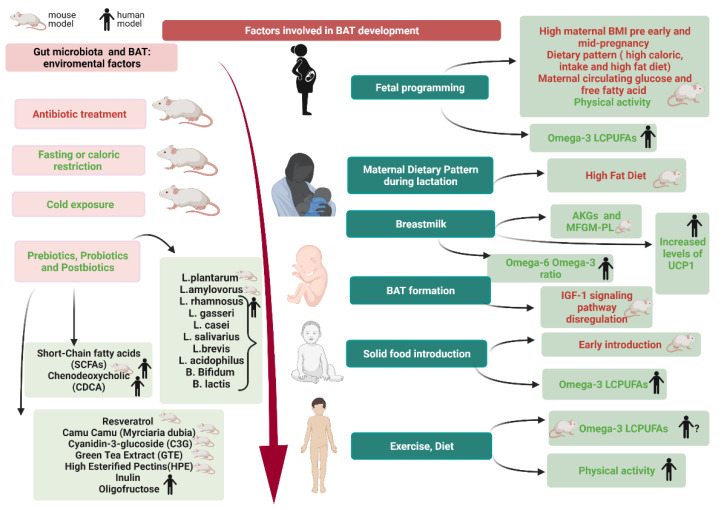
Factor involved in (brown adipose tissue) BAT development. In humans, modulation of BAT occurs during gestation and continues during lactation through lifestyle and maternal BMI. A preterm or low birth weight babies may have a dysregulation of the IGF-1 pathway. Breast milk rich in AGKs and MFGM-PL appears to promote BAT. An early introduction of solid food (before 4-month-old age) is related to a lower activity of the sympathetic nervous system on BAT. Later, in childhood and adolescence exercise and an intake of omega-3 seems to promote BAT development. Gut microbiota can be modulated by environmental factors that appear to promote BAT and its activation (cold exposure, fasting or caloric restriction, prebiotics, probiotics, postbiotics). Made by © BioRender 2021.

## Data Availability

Not applicable.

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
