# Peer review of "Brown Adipose Tissue: New Challenges for Prevention of Childhood Obesity. A Narrative Review"

_nutrients, 2021, doi:10.3390/nu13051450_

Round 1
Reviewer 1 Report
The authors have comprehensively reviewed the literature on brown adipose tissue in the context of the prevention of childhood obesity. Minor concerns are listed below.
- The authors did not explain beige adipose tissue and how it differs from BAT. A short paragraph on this topic is suggested.
- Although authors make comments on the side effects associated with the activation of BAT, a separate section on the limitations of activating BAT will make the review article wholesome.
Author Response
Milano, April 14, 2021
Object: Manuscript minor revisions (nutrients-1184124)
Dear Editors,
On behalf of my co-authors, I wish to submit the revised version of our manuscript entitled: “Brown adipose tissue: new challenges for prevention of childhood obesity. A narrative review.” (nutrients-1184124) for consideration for publication in Your “Newborn Nutrition" special issue. We appreciated the positive feedback, and we have made all the required adjustments, which are highlighted in yellow within the text. The suggestions are discussed here below in a point-by-point fashion.
We look forward to hearing Your response,
Kind regards,
Elvira Verduci
Please address all correspondence to:
Prof. Elvira Verduci, MD, PhD
Associate Professor of Pediatrics, Department of Pediatrics, Vittore Buzzi Children’s Hospital, University of Milano, Milano, Italy. E-mail: elvira.verduci@unimi.it
Dr. Stephana Carelli, PhD
Department of Biomedical and Clinical Sciences "L. Sacco", University of Milano, Milano, Italy.
Pediatric Clinical Research Center Fondazione Romeo ed Enrica Invernizzi, University of Milano, Milano, Italy. E-mail: stephana.carelli@unimi.it
Reviewer #1:
The authors have comprehensively reviewed the literature on brown adipose tissue in the context of the prevention of childhood obesity.
R: We thank the reviewer for the positive feedback and interesting suggestions.
Minor concerns are listed below.
- The authors did not explain beige adipose tissue and how it differs from BAT. A short paragraph on this topic is suggested.
R: We have added a paragraph on this topic as suggested. Please see the new paragraph 3.1 entitled “Brite or beige adipocytes” at pages 4 and 5.
(3.1 Brite or beige adipocytes
Recent data suggests the existence of another adipose cell phenotype, which shows both white and brown adipose cells features and is therefore called brite or beige adipose tissue. Beige adipocytes are usually located into WAT sites, especially in the subcutaneous WAT depots, and the phenomenon of their appearance in WAT depots is referred as “browning” [9,11,12].
Beige cells resemble brown fat cells in having multilocular lipid droplets and capacity for UCP1-mediated thermogenesis, however at the basal expression of UCP1 is very low [8]. In fact, the thermogenic feature of beige adipose tissue (BeAT) appears under prolonged cold exposure or as consequence of chronic β-adrenergic stimulation [9,11,12]. Under these conditions, the pre-existing beige adipocytes (which may appear unilocular in the basal state) will go through phenotypic trans-differentiation, and browning will appear both morphologically and histochemically [11]. Once stimulated beige cells activate the expression of UCP1 at very comparable levels to those of the classic brown fat cells [11]. Thus, the beige cells have the capability to switch between an energy storage and energy dissipation phenotype.
The adipogenesis of white and brown adipocytes includes the development of pre-adipocytes from mesenchymal stem cells (Myf5-negative cells or Myf5-positive cells) that further differentiate to mature adipocytes (white adipocyte or brown adipocyte respectively) [7]. Regarding the beige cells, their origins remain a matter of debate. Beige-type cells may generate from white-to-brown adipocyte trans-differentiation or transformation of white adipocytes [12]. Alternatively, it has been hypothesized that these cells origin from the de novo differentiation of a distinct sub-population of WAT progenitors (presenting CD137 and TMEM26 as surface markers) which can give rise to either white or beige adipocytes depending upon the stimuli [7,12]. Browning or “beiging” takes place in response to a variety of external stimuli such as chronic cold exposure, cancer cachexia, caloric restriction, exercise and also bariatric surgery [8]. Once activated, beige adipocytes secrete multiple autocrine and paracrine factors that control the expansion and activity of BAT and the extent of browning of white adipose tissue [9]. Therefore, BeAT development and functions go along with BAT activity in the control and prevention of obesity onset.).
- Although authors make comments on the side effects associated with the activation of BAT, a separate section on the limitations of activating BAT will make the review article wholesome.
R: Thank you. As suggested, we have inserted a separate section concerning the limitations of BAT activation. Please see the new paragraph 6.5 entitled “Limitations of BAT activations” at pages 15 and 16.
(6.5 Limitations of BAT activation
The limits of BAT activation, in conclusion, are represented by environmental factors but also by physiological factors. It has been reported how some characteristics of the mother during breastfeeding may limit the activation of BAT. These are represented by the nutritional status of the mother [25], her diet excessively rich in fat during breastfeeding [28,29], and the type of breastfeeding. Formula milk does not contain AGKs that promote the activation of BAT [111,112] and, moreover, a too early complementary feeding shows a reduction of the vivo BAT sympathetic nervous system activity and may represent factors that limit the activation of BAT [122]. In addition, a central role is given to the gut-microbiota since the use of antibiotics generating a microbial depletion limits BAT activation [132]. Environmental factors such as the continuous exposure to mild temperatures determines a lower activation of BAT with consequences on glucose homeostasis [91]. Moreover, specific physiological factors reduce the activation of BAT and these are essentially represented by the state of maturation of the organism. During the growth of the child the amount of BAT present in the organism decreases. In childhood BAT is present in greater quantities in the early stages of life, promoting NST, then it begins to decrease, but it plays an important role in the development of the skeletal muscle system and later in adulthood it is involved in lipid and glucose metabolic processes [72]. Therefore, the “maturation” of the body, physiologically determines the decrease of BAT amount limiting its activation. Consequently, if BAT activation is properly promoted in childhood, it would allow the body to face adulthood with structural and metabolic functions at their best.).
Reviewer 2 Report
Please find the attached file below.

Author Response
Milano, April 14, 2021
Object: Manuscript minor revisions (nutrients-1184124)
Dear Editors,
On behalf of my co-authors, I wish to submit the revised version of our manuscript entitled: “Brown adipose tissue: new challenges for prevention of childhood obesity. A narrative review.” (nutrients-1184124) for consideration for publication in Your “Newborn Nutrition" special issue. We appreciated the positive feedback, and we have made all the required adjustments, which are highlighted in yellow within the text. The suggestions are discussed here below in a point-by-point fashion.
We look forward to hearing Your response,
Kind regards,
Elvira Verduci
Please address all correspondence to:
Prof. Elvira Verduci, MD, PhD
Associate Professor of Pediatrics, Department of Pediatrics, Vittore Buzzi Children’s Hospital, University of Milano, Milano, Italy. E-mail: elvira.verduci@unimi.it
Dr. Stephana Carelli, PhD
Department of Biomedical and Clinical Sciences "L. Sacco", University of Milano, Milano, Italy.
Pediatric Clinical Research Center Fondazione Romeo ed Enrica Invernizzi, University of Milano, Milano, Italy. E-mail: stephana.carelli@unimi.it
Reviewer #2:
The authors review the exciting research field of childhood obesity by focusing on the potential role of BAT development and function in terms of fetal programming via maternal diet especially in the modulation of microbiota. This review manuscript is very comprehensively and precisely described. There is no specific comment except for a typographical error in page 12. The subtitle “6.2.3. Postbiotics” should be corrected to “6.4.3. Postbiotics”.
R: We wish to thank the reviewer for the positive feedback. We have corrected the subtitle as suggested. Please see the correct numbering of paragraph 6.4.3 at page 14.